# Is There a Relationship between Psychotic Disorders and the Radicalization Process? A Systematic Review

**DOI:** 10.3390/medicina60060926

**Published:** 2024-06-01

**Authors:** Pierluigi Catapano, Salvatore Cipolla, Corrado De Rosa, Stefania Milano, Daniela Vozza, Davide Guadagno, Francesco Perris, Gaia Sampogna, Andrea Fiorillo

**Affiliations:** Department of Psychiatry, University of Campania “Luigi Vanvitelli”, 80138 Naples, Italy; pierluigi.catapano@unicampania.it (P.C.); salvatore.cipolla@unicampania.it (S.C.); andrea.fiorillo@unicampania.it (A.F.)

**Keywords:** radicalization, psychosis, mental health, terrorism risk assessment, lone actors, mass shooters

## Abstract

*Background and Objectives*: Radicalization, a complex and multifaceted phenomenon, has been a subject of increasing concern in recent years, particularly due to its potential connection to acts of mass violence and terrorism. This systematic review examines the intricate link between radicalization and psychotic disorders, utilizing various sources such as observational studies, case reports, and series. It aims to highlight the prevalence of schizophrenia spectrum and other psychotic disorders among radicalized individuals and to define the role of mental health professionals in dealing with this issue, contributing to the development of prevention and treatment strategies. *Materials and Methods*: The methodology involved an extensive literature search across PubMed, Scopus, and APA PsycINFO up to 1 February 2024, adhering to PRISMA guidelines. The study focused on radicalization and psychotic disorders as defined by DSM-5 criteria, excluding other mental disorders. A population sample of 41 radicalized individuals diagnosed with psychotic disorders was selected, among which schizophrenia was identified as the predominant condition. *Results*: It was observed that 24% of these individuals passed away soon after committing their crimes, leading the researchers to rely on retrospective data for their diagnoses. The use of diverse assessment tools for psychiatric diagnosis and the lack of a standardized method for diagnosing or assessing involvement in the radicalization process were also noted. Despite limitations like reliance on observational studies and case reports, which result in low evidence quality and varied methodologies, our work provides a valuable contribution to clarifying the relationship between radicalization and psychotic disorders. However, further clinical studies are needed to delve deeper into these aspects. *Conclusions*: In conclusion, our review points out that individuals with psychotic disorders do not have a higher crime rate than the general population and warns against associating crimes with mental illness due to the stigma it creates. The lack of uniform psychiatric diagnostic tools and radicalization assessment highlights the need for more standardized risk assessment tools and validated scales in psychiatric diagnosis to better understand the relationship between radicalization and psychotic disorders and to develop integrated protocols.

## 1. Introduction

Radicalization is a complex process in which people become motivated to use violent acts against members of other groups or specific targets in order to achieve social or political changes [1,2]. Being a complex phenomenon, many theories, stages, and factors have been proposed [3,4]. According to Misiak et al. (2019), radicalization is a process by which individuals adopt political, social, and religious ideation that leads to the initiation of mass violence acts [5], while Shmid et al. (2013) define radicalization as “both an individual and group process, whereby political actors and groups that are politically polarized renounce dialogue, agreement, and tolerance and use either non-violent pressure and coercion, or various forms of political violence including violent extremism (terrorism and war crimes)” [6]. The European Commission defines violent radicalization as the phenomenon of people embracing opinions, viewpoints, and ideas that might lead to terrorist acts [7].

Radicalization is a complex process, in which factors such as depressive symptoms, suicidal thoughts, feelings of unfairness or disgrace, issues of identity and belonging, existential fragility, and the search for meaning can play a significant role [8]. In fact, the ideology proposed by a radicalized group provides a theoretical framework for managing all these challenges. Different theoretical models have been proposed for explaining the radicalization process, such as the four stages model by Borum (2003) [9] and the Staircase model by Fathali M. Moghaddam (2005) [10]. Borum’s model describes a psychological sequence from detecting an unwanted event to dehumanizing the perpetrators, seen as responsible, which results in violence. The Staircase model identifies five stages of psychological justifications or psychological interpretations up to the level of terrorism. However, these models have limitations since they tend to underestimate the role of ideology, ignore the non-linearity and reversibility of radicalization, and neglect the discrepancy between attitudes and behaviors [11].

Based on these limitations, more sophisticated models have been proposed, including the pyramid and the two-pyramid model. According to the former (McCauley and Moskalenko, 2008), radical ideology can be common in the general population (those defined as “sympathizers”), but a few individuals located at the top of the pyramid (defined as “terrorists”) commit violent acts [12]. Recently, this model has been revised by the two-pyramid model, which includes a variety of subjects from those neutral (defined as “inert” individuals located at the basis) to those compelled to act violently, located at the apex of the pyramid [13]. An alternative dynamic model is the Attitudes–Behaviors Corrective (ABC) model of violent extremism, considering two dimensions: the “behaviors” axis categorizes roles from suicide attackers to support roles, while the “attitudes” axis reflects the ideological justification for violence. Both models offer a dynamic vision of the radicalization process and can be considered an alternative to each other [14]. A schematic representation of the main models of radicalization is reported in Figure 1.

Ideology remains an intrinsic factor in the radicalization process, but it often begins with individuals dissatisfied with their lives, society, or the local and foreign policies of their governments. There is no specific vulnerability profile, but those who belong to groups on the margins of society and experience discrimination or loss of identity are easy prey for radicalizing recruitment [7].

A complex relationship has been proposed between the radicalization process and the presence of mental disorders in perpetrators [15,16,17], which can be useful to study in order to identify possible protective and risk factors [5]. However, available data are controversial, and no mental disorder can be considered as a specific risk factor for radicalization [18,19,20].

It has been hypothesized that mental health characteristics might be associated with a risk of radicalization [21,22,23]. However, a recent systematic review concluded that caution should be taken on how the association between “mental health” and “radicalization” is being claimed, because of limited evidence so far, and a number of methodological limitations of studies addressing this issue [24,25,26]. There is some evidence that lone actors might represent a specific subgroup of subjects with extreme beliefs, which can be characterized by a high prevalence of psychotic and/or mood disorders. Furthermore, the risk of malingering in criminals and radicalized people has been highlighted in order to obtain legal benefits [27]. It is therefore relevant to identify the specific clinical features separating radical ideologies from delusional thoughts [28,29,30,31,32,33].

Based on these premises, the present systematic review aims to (1) provide prevalence rates of schizophrenia spectrum and other psychotic disorders among radicalized individuals; (2) identify assessment tools for identifying psychotic symptoms/syndrome in radicalized people; and (3) clarify the role of mental health professionals in the prevention and management of radicalization.

## 2. Materials and Methods

### 2.1. Search Strategy

An extensive literature search for relevant articles was performed from inception up to 1 February 2024. The following search keys: (“Mental Disorders”[Mesh]) NOT (“Obsessive Compulsive Disorder”[Mesh])) NOT (“Stress Disorders, Post-Traumatic”[Mesh])) NOT (“Anxiety Disorders”[Mesh])) AND ((radicalization) OR (radicalization) were entered on PubMed; (“Mental Disorders”) NOT (“Stress Disorders, Post-Traumatic”)) NOT (“Obsessive Compulsive Disorder”)) NOT (“Anxiety”)) AND ((“Terrorism”) OR ((radicalization) OR (radicalization) were entered on Scopus and APA PsychInfo. The reference lists of the included articles were screened to identify additional relevant studies.

The search method was used according to the Preferred Reporting Items for Systematic Review and Meta-Analysis (PRISMA) statement, as applicable [34]. The protocol of this systematic review has not been registered in any dataset.

### 2.2. Selection Criteria

Papers were selected for the review if they were (1) clinical observational or interventional studies about radicalization and people suffering from schizophrenia spectrum and other psychotic disorders, as defined by DSM-5 criteria [35], even if this was not the primary outcome of the selected study; (2) case reports and case series concerning radicalized individuals affected by schizophrenia spectrum and other psychotic disorders; (3) studies that considered individuals as radicalized if they were involved in terrorism, mass violence, mass shooting; “lone wolves” and “lone actors” were also considered as radicalized people; (4) written in English.

Studies focused on other mental disorders, such as affective disorders without psychotic symptoms, anxiety disorders, post-traumatic disorders, and obsessive–compulsive disorder, were excluded. Non-original papers, such as systematic reviews, meta-analyses, and narrative reviews, were also excluded.

### 2.3. Selection Process

D.V., D.G., and S.M. extracted the relevant data and synthesized them in a tabular format; G.S. triple-checked the extracted data for accuracy. Inter-rater reliability, referring to the degree of agreement between researchers, was calculated, with a Cohen’s kappa score of 0.9.

### 2.4. Population Sample Identification

Starting from a broad initial population of radicalized individuals, our research first identified those with a diagnosed mental disorder. We then narrowed our focus to individuals with schizophrenia and other psychotic spectrum disorders, including cases where the diagnosis was established post-mortem via psychological autopsy.

### 2.5. Risk of Bias Assessment

Two authors (P.C. and S.C.) independently assessed each selected study for risk of bias using the ROBINS-E tool [36], which represents a structured method for evaluating the risk of bias in observational studies. The overall risk of bias was rated from “low risk of bias, except for concerns about uncontrolled confounding” to “high”. Only one study was deemed to have an overall low risk of bias [37]. Appendix A details the domains and subdomains considered. Disagreements were resolved through discussion among the two authors or by consulting a third author (G.S.). For case reports and case series, all chosen articles were evaluated using the Joanna Briggs Institute (JBI) Critical Appraisal tools for Case Reports [38], based on the CARE guidelines [34,39] and the JBI Critical Appraisal Checklist for Case Series [40].

## 3. Results

Based on our search strategy, 1004 papers were identified, 132 of which were duplicates and were removed. Among the remaining articles (N = 872), 149 articles were excluded by reading titles and abstracts, and 723 papers were evaluated as full-text. Eight studies matched the inclusion criteria and were included in the review, while two additional papers were included by snowballing and citation checking [41,42]. The selection process is summarized in Figure 2.

The final sample included ten studies, comprising observational studies (N = 5), case reports (N = 4) [42,43,44,45], and case series (N = 1) [46]. As regards study design, two were retrospective studies [41,47], one was a case–control study [32,37], one prospective longitudinal study [48], and one was a cross-sectional study [49]. Details on clinical studies are reported in Table 1. Table 2 shows details of included case reports and case series. Two observational studies examined the same subjects’ population [41,47]. Observational studies can provide essential data about the available evidence on the prevalence rates of psychotic disorders in radicalized people. Longitudinal studies can inform about the likelihood of individuals with psychotic disorders becoming radicalized or committing violent acts over time. Case reports and case series may offer in-depth information about the specific factors involved in the radicalization process of those with a diagnosis of psychotic disorder.

The sample size ranged from 1 single person [42,43,44,45] to 55 participants [41,47], for a total of 128 individuals identified as radicalized, and they were mainly male.

Our analysis focused on radicalized people affected by psychotic disorders, namely 41 patients out of the 128 radicalized individuals. Despite originating from a mixed population, we were able to isolate this specific subgroup and extract valuable data for our study. The available data did not allow us to provide any gender-based description.

Considering the subsample of radicalized people with a psychotic disorder (N = 41), in 70.7% of cases (N = 29), patients were affected by schizophrenia [44,45,47,49]. In 7.3% of cases (N = 3), patients suffered from chronic delusional disorder [47,49]. In 4.9% of cases (N = 2), patients were affected by a brief psychotic episode [37]. In 12.2% (N = 5) patients suffered from a psychotic disorder not otherwise specified [42,43,46,48], and in 4.9% (N = 2), subjects experienced psychotic disorder at the time of assessment despite the primary diagnosis being borderline personality disorder or not having been established [41,47].

In 24% of cases (N = 10 patients), retrospective data were used in order to formulate a diagnosis using the technique of the psychological autopsy [41,42,45,47]. In these cases, the causes of death were suicide or being killed by police officers.

Different tools were used to perform the psychiatric assessment (Table 3). The clinical evaluation based on the available psychiatric and forensic documentation was used in case reports and case series [42,43,44,45,46] and in one observational study [48], without any validated assessment tools. In the remaining studies, the following assessment tools were used: an ad hoc semi-structured interview; the Adverse Childhood Experience (ACE) Questionnaire; the Mini International Neuropsychiatric Interview (MINI-KID 2) (a structured interview for children and adolescents modified by authors of cited study adding the subscale of the MINI for adults for psychotic disorders); the Abbreviated Diagnostic Interview for Borderline (Ab-DIB); Adolescent Depression Rating Scale (ADRS); Columbia Evaluation Scale for Suicidal Risk Gravity (C-SSRS); Beck Hopelessness Scale (BHS) [37]; Wechsler Adult Intelligence Scale (WAIS III); Mini International Neuropsychiatric Interview (MINI); the Symptom Checklist 90 (SCRL 90); the Young Schema Questionnaire (YSQ-L2/SQ2); Rorschach’s psychodiagnostic projective tests, thematic apperception test (TAT) [44]; and MINI and an ad hoc 62-item questionnaire in studies by Cerfolio et al. (2022) and Glik et al. (2022) [41,47].

As regards the radicalization process, in 78% of cases (N = 32), perpetrators were mass shooters [41,42,43,47]; in 12,2% (N = 5), they were lone wolf terrorists [44,45,46,49]; in 4.9% (N = 2), they were members of terroristic groups [32,37], and the remaining 4.9% of cases (N = 2) were unspecified [48].

The radicalization process was evaluated using a validated scale (Terrorist Radicalization Assessment, TRAP-18) only in one study [45], while the risk of committing unspecified violent acts was evaluated by using Historical Clinical Risk-20 (HCR-20) [48,50].

The prevalence of schizophrenia spectrum disorders significantly varied, ranging from 7.4% in studies by Garcet (2021) [49] to 54.5% in the studies by Cerfolio et al. (2022) [47] and Glick et al. (2021) [41].

Garcet et al. (2021) [49] proposed a model of cognitive and affective transformation of self-definition and sense construction in violent radical engagement and found a prevalence of mental disorders of 7.4% in radicalized individuals (N = 2 individuals out of 27). Bronsard et al. (2022) [37] found the same rate of lifetime psychotic episodes in a sample of radicalized and non-radicalized people engaged in criminal activities (13.3% vs. 13%, respectively). Morris and Meloy (2020) analyzed the association between mental disorders and involvement in terrorist activities, concluding that although there is a high prevalence of mental illnesses in these subjects, having a mental disorder does not mean a higher risk of violent terrorist acts [48].

Cerfolio et al. (2022) highlighted that many radicalized individuals do not receive any diagnosis of mental disorder at the time of the legal process, despite the presence of a mental disorder being indicated by the available clinical information, while Glick et al. (2022) outlined that the diagnosis was often wrong. In some cases, it was not possible to assess the definitive diagnosis due to the lack of medical information to verify if patients met diagnostic criteria. Both studies concluded that subjects did not receive adequate psychiatric care [41,47].

## 4. Discussion

The relationship between radicalization and mental disorders is very complex, as radicalization is a multidimensional construct. Few studies have been conducted so far with a specific focus on the role of psychotic spectrum disorders on the radicalization process and the risk of committing criminal acts [51,52,53,54,55,56].

The first relevant finding of our systematic review is the extreme heterogeneity in prevalence rates of schizophrenia spectrum disorders in samples of radicalized people. These data deserve careful evaluation because they can be due to the lack of consistency in the type of assessment tools adopted or to the lack of specific measures to be used in people committing terroristic crimes. Moreover, the extreme variability in prevalence rates also highlights the complex association between the process of radicalization and the potential role of mental disorders. In line with this, Bronsard et al. [37] reported the same prevalence rate of lifetime psychotic episodes in radicalized and non-radicalized individuals.

Another interesting finding is the high prevalence of psychotic disorders in the subsample of terrorists acting as lone wolf/mass shooters [47,51], confirming data from previous studies [25,57]. It is essential to differentiate between people affected by mental disorders and committing terroristic attacks and other radicalized people, since it should be noted that radicalized terroristic groups may be reluctant to recruit individuals with mental disorders [25]. In fact, they may even stigmatize people with mental problems, thinking that they are unreliable, hard to train, and difficult to coordinate [58], confirming stereotypes of the general population about people with severe mental disorders [59]. In the longitudinal Prevent program, a European strategy focusing on early intervention for individuals at risk of terrorist activities, Morris and Meloy) [48] found that 22% of a radicalized group was diagnosed with unspecified psychotic disorders. Such a relatively high prevalence rate of psychosis in this group highlights the need to understand the relationship between mental disorders and radicalization.

It should be also noted that people affected by psychotic spectrum disorders, especially those suffering from delusional disorder, could be more easily influenced by terrorist propaganda. Delusional ideas about terrorist group membership or perceived threats from a hostile ‘enemy’ can increase vulnerability and the risk associated with violence. Similarly, persecutory and paranoid delusions create a sense of threat, while grandiose or referential content can lead individuals to believe they have an important role in combating threats, potentially through terroristic attacks. Somatic delusions may cause individuals to feel physically harmed by a group. Their intensity and nature, the distress they cause, and the subjects they blame are very important in assessing vulnerability and potential targets of violence [60]. A possible risk is that of attributing violent/criminal behaviors to people suffering from psychosis as a consequence of stereotypes, linking violent behaviors with mental disorders [61].

Given the complexity of the relationship between radicalization and mental health problems, it should be highlighted that data are still controversial on the higher prevalence of terroristic acts committed by people with mental disorders compared to those without mental disorders. Whether mental disorder has some relevance, it is not causal and interacts with a myriad of political, social, environmental, situational, and biological factors [58].

For example, the relationship between personality traits or disorders and extreme violence appears fleeting due to limitations in research design, although borderline, narcissistic, and antisocial personality disorders are the most common among terrorists [62]. Other scholars evaluated the link between specific mental illnesses, especially depression, and the radicalization risk, but research generally did not confirm this relationship [58].

Other interesting data are related to the lack of standardized assessment procedures/tools, as confirmed by the fact that in 60% of considered papers, psychiatric diagnosis was based on retrospective clinical assessment, and data were obtained from interviews, personal writings, expert reports, and court records [42,43,44,45,46,48]. Furthermore, another source of bias is that, in many cases, the diagnosis was confirmed post-mortem [63,64]. The adoption of standardized assessment tools should represent a reasonable strategy for improving the quality of data on the complex relationship between mental disorders and radicalization and for improving the external validity of such data.

As regards the evaluation of the risk of radicalization, although several tools exist [65], they are only rarely used [48]. However, these instruments are essential for the early detection of people at risk for violent radicalization and for developing effective tertiary prevention programs [66,67,68]. De-radicalization is a long journey that needs multilevel and integrated interventions, with the identification of innovative places—different from traditional jails—in which people should be admitted [69].

In the complex framework of activities to be promoted for the de-radicalization process, mental health professionals should contribute to disentangling the complex relationship between mental disorders and radicalization.

The present study has some limitations that must be acknowledged. Firstly, the selected studies were observational studies and case reports/case series. Therefore, the quality of the evidence is relatively low. Moreover, the selected studies were limited by a high risk of bias. Another limitation is due to the extreme heterogeneity of adopted tools. Thirdly, the research methodologies varied across studies, especially in terms of the assessment tools. Moreover, regarding the assessment tools adopted in the papers, it is notable that only in one study was the MINI Neuropsychiatric Interview used, which provides an objective way to diagnose psychotic disorders. Finally, due to such heterogeneity, it was not possible to perform a meta-analysis.

Another limitation is the limited final sample of studies included (observational = 5; case reports = 4 and case series = 1). This seems like a very limited number of studies for a review paper, especially given the types of studies chosen for the final review.

Despite these limitations, the present work should contribute to understanding the mechanisms linking radicalization and psychotic disorders.

## 5. Conclusions

According to our systematic review, a clear correlation between the radicalization process and having a psychotic disorder could not be identified. Moreover, assessment tools used to make psychiatric diagnoses were highly heterogeneous. Finally, the degree of radicalization was evaluated in one case report only [45].

In conclusion, the training of mental health professionals on this specific topic should be improved, and assessment tools should be developed for evaluating the risk of the radicalization process [25,70,71] and then particularly used in countries where the risk of radicalization is higher [72,73,74].

## Figures and Tables

**Figure 1 medicina-60-00926-f001:**
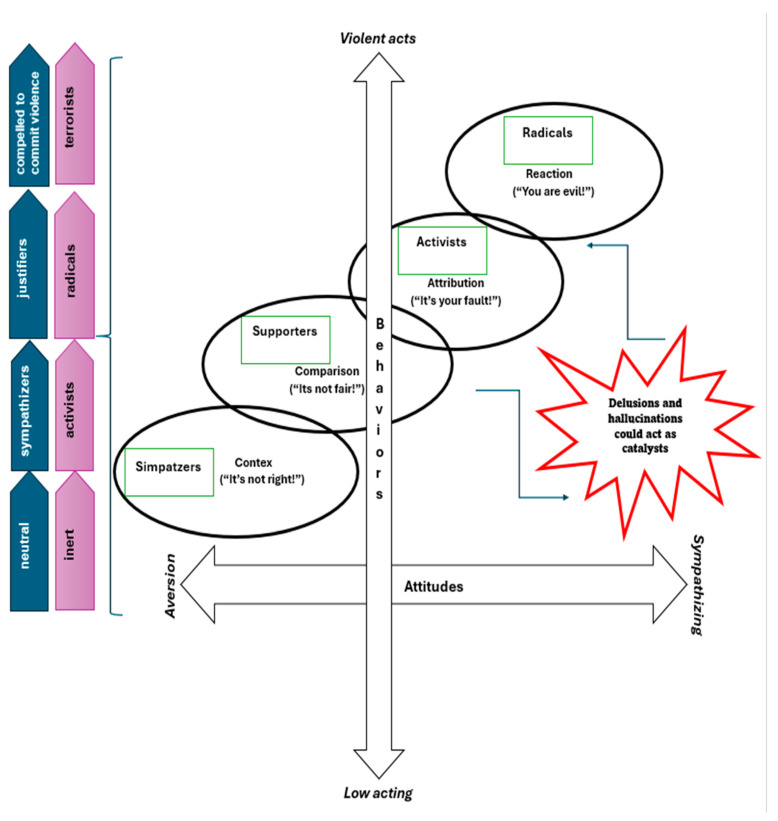
Flowchart of the theoretical process leading to radicalization, based on a mixture of most validated models.

**Figure 2 medicina-60-00926-f002:**
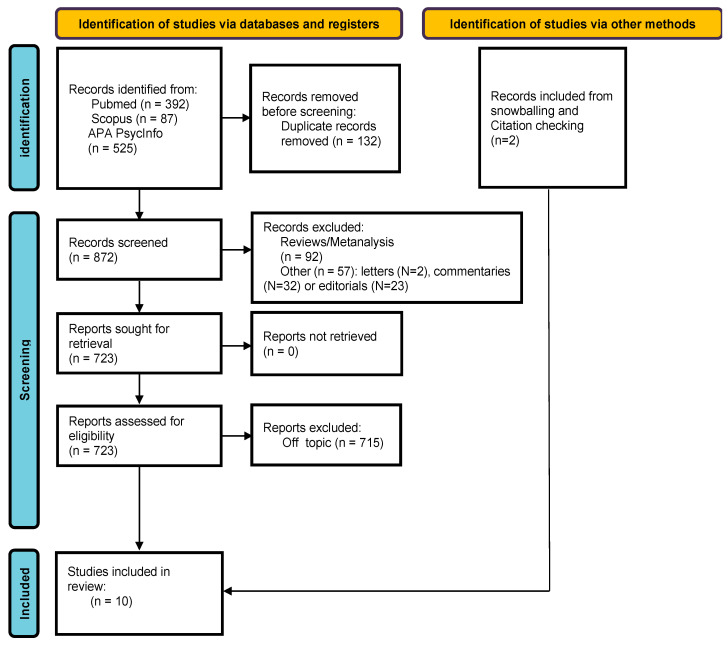
Flowchart of included studies.

**Table 1 medicina-60-00926-t001:** Main characteristics of the included clinical studies (N = 5).

Author, Country (Year of Publication)	Type of Study	Sample	Psychiatric Assessment	Main Results	Conclusions
Bronsard G., France (2022) [37]	Case–control study	N = 15 adolescents involved in terrorist activities vs. N = 101 teenagers convicted for non-terrorist delinquency who were placed in CEC.	Ab-DIB; ACE questionnaire; ADRS; BHS; C-SSRS; MINI-KID 2 (modified with the subscale of the MINI for adults for psychotic disorders); Semistructured interviews.	In the radicalized group: the psychiatric examinations did not show axis 1 disorders for most of them (N = 10, 66.7%). For the other five adolescents, comorbid associations with the following diagnoses were found: BPD, GAD, CD, and Depression. Psychotic lifetime episodes were found in 2 cases (13.3%).	Among adolescents involved in terrorist activities, only two of them reported psychotic lifetime episodes.
Cerfolio N., United States (2022) [47]	Retrospective Observational study	N = 55 persons identified as committing a mass shooting in the USA between January 1982 and September 2019 Participants were grouped in those surviving after the crime (N = 35) and those dead (N = 20)	A 62-item questionnaire to compile the data collection from multiple sources and record the psychiatric assessments using DSM-5 criteria; MINI	In the subgroup of the 35 assailants who survived, 28 had a primary DSM-5 diagnosis: 18 (56%) had SCZ; 3 had BD-I; 2 had delusional disorder (persecutory); 2 had personality disorders (1 PPD, 1 BPD with current psychotic disorder); 2 had SRD, and 1 had PTSD. In addition, in 4 cases with no primary diagnosis, 1 case was found with a current psychotic disorder. In the subgroup of 20 criminals who died after the event, authors were able to provide a diagnosis only in 8 cases (SCZ).	Many perpetrators did not seek psychiatric care due to the stigma against mental illness. Even when they did engage with a healthcare provider prior to their violence, they were often misdiagnosed.
Garcet S., Belgium (2021) [49]	Cross-sectional Observational study	N = 27 radicalized individuals, Aged 17–31 years (mean: 24.6 years).	MINI; Rorschach’s psychodiagnostic projective tests; SCL-R90; TAT; WAIS III for Adult; YSQ-L2/SQ2.	Schizoid personality (N = 1), SCZ (N = 1), and delusional disorder (N = 1) were observed. Most subjects had a comorbidity with personality disorders.	A model of cognitive and affective transformation of self-definition and sense construction in violent radical engagement has been created to study how radicalized individuals process and integrate information at different societal levels into their own belief systems, thereby improving the comprehension of these cognitive integration mechanisms.
Glick I., United States (2021) [41]	Retrospective Observational study	N = 55 persons identified as committing a mass shooting in the USA between January 1982 and September 2019 Participants were grouped in those surviving after the crime (N = 35) and those dead (N = 20)	A 62-item questionnaire to compile the data collection from multiple sources and record the psychiatric assessments using DSM-5 criteria; MINI	In the subgroup of the 35 assailants who survived, 28 had a primary DSM-5 diagnosis: 18 (56%) had SCZ; 3 had BD-I; 2 had delusional disorder (persecutory); 2 had personality disorders (1 PPD, 1 BPD with current psychotic disorder); 2 had SRD, and 1 had PTSD. In addition, in 4 cases with no primary diagnosis, 1 case was found with a current psychotic disorder. In the subgroup of 20 criminals who died after the event, authors were able to provide a diagnosis only in 8 cases (SCZ).	A significant proportion of mass shooters experienced unmedicated and untreated psychiatric disorders.
Morris A., United Kingdom (2020) [48]	Prospective observational study	N = 23 individuals referred to prevent elements of U.K. national counterterrorism strategy	Analysis of records about criminal and psychiatric history; HCR-20.	N = 2 participants had a diagnosis of psychotic disorder (22%). In many cases, participants received more than one diagnosis.	Those flagged as vulnerable to involvement in terrorist activities have a high prevalence of mental disorders. It does not indicate that individuals with mental disorders are at higher risk of terrorist violence.

AB-DIB = Abbreviated Diagnostic Interview for Borderline; ACE = adverse childhood experience; ADRS = Adolescent Depression Rating Scale; AMT = criminal association to commit terrorism; BD-I = bipolar disorder Type I; BHS = Beck Hopelessness Scale; BPD = borderline personality disorder; CD = conduct disorder; CEC = closed educational centers; C-SSRS = Columbia Evaluation Scale for Suicidal Risk Gravity; DSM-5: Diagnostical and Statistical Manual of Mental Disorders (5th edition); DSPD = dissocial personality disorder; EUPD = emotionally unstable borderline personality disorder; GAD = generalized anxiety disorder; HCR-20 = Historical Clinical Risk-20; MINI = Mini International Neuropsychiatric Interview; PD = Personality Disorder; PPD = Paranoid Personality Disorder; PTSD = Post traumatic stress disorder; SCL-R90 = the Symptom Checklist 90; SCZ = schizophrenia; SRD = substance related disorder; TAT = thematic apperception test; WAIS = Wechsler Adult Intelligence Scale; YSQ-L2/SQ2 = The Young Schema Questionnaire.

**Table 2 medicina-60-00926-t002:** Case report and case series.

First Author, Country (Year of Publication)	Type of Study	Sample Characteristics	Main Diagnosis	Psychiatric Assessment	Conclusions and Results
Amador X., United States (2000) [44]	Case report	Lone wolf terrorist Theodore Kaczynski	Schizophrenia	Analysis of T. Kacztnski’s psychiatric history	This case report is in agreement with recent research that has indicated that the etiology of poor insight is not only due to defense mechanisms but also linked to neurological deficits such as anosognosia.
Cotti P., France (2015) [43]	Case report	Mass shooter Anders Breivik	Not specified psychotic disorder	Analysis of Breivik’s words and writings	It would seem that the sociocultural and political determinants of ideological discourses can lead individuals who do not have a marked psychopathological character to topple over into martyrdom.
Cotti P. and Meloy J.R., United States (2019) [42]	Case report	Lone actor terrorist Tamerlan Tsarnaev	Not specified psychotic disorder	Analysis of various sources about Tamerlan Tsarnaev's history	The Tsarnaev case allowed researchers to give a specific example of the dynamic of the nexus of psychopathology and ideology, a key dynamic in the mentally disordered lone actor’s radicalization process.
Kupper J., United States (2023) [45]	Case report	Lone wolf terrorist Tobias Rathjen	Schizophrenia	Qualitative analysis of primary and secondary sources. TRAP-18 was also used.	The complex interplay of his delusions, obsessions, and extremely overvalued beliefs that drove his fixation reveal the difficulty of clinically understanding such a case, and also the necessity of doing so, to attempt to risk mitigate these types of subjects by threat assessment teams.
Spaaij R., Australia (2010) [46]	Case-series	5 cases of lone wolves: David Copeland, Franz Fuchs, Theodore Kaczynski, Volkert Van der Graaf, and Yigal Amir	Only for David Copeland symptoms and history of not specified psychotic disorder are reported.	Documentary analysis of media reports, literature, documents and letters written by the perpetrators, police and court transcripts, and psychological and/or psychiatric evaluations.	The likelihood of lone wolf terrorists suffering from psychological disturbances is relatively high.

TRAP-18 = Terrorist Radicalization Assessment Protocol.

**Table 3 medicina-60-00926-t003:** Assessment tools used in each included study.

		Clinical Studies					Case Report			Case Series	
		Bronsard G. (2022) [37]	Cerfolio N. (2022) [47]	Garcet S. (2021) [49]	Glick I., (2021) [41]	Morris A. (2020) [48]	Amador X. (2000) [44]	Cotti P. (2015) [43]	Cotti P. and Meloy J.R. (2019) [42]	Kupper J. (2023) [45]	Spaaij R. (2010) [46]
	Clinical Assessment										
**Main Diagnosis**	*Ad hoc semi-structured interview*	+									
	*Ad hoc 62-item questionnaire based on DSM V criteria*		+		+						
	*Clinical judgment based on the analysis of various anamnesis sources*					+	+	+	+	+	+
	*Mini International Neuropsychiatric Interview (MINI)*		+	+	+						
	*Mini International Neuropsychiatric Interview (MINI- KID 2)* *	+									
	*Symptom Checklist 90 (SCL-R90)*			+							
**Depression**	*Adolescent Depression Rating Scale (ADRS)*	+									
	*Beck Hopelessness Scale (BHS)*	+									
	*Columbia evaluation scale for suicidal risk gravity (C-SSRS)*	+									
**Personality**	*Abbreviated Diagnostic Interview for Borderline (Ab-DIB)*	+									
	*Adverse Childhood Experience (ACE) Questionnaire*	+									
	*Thematic Apperception Test (TAT)*			+							
	*Rorschach’s psychodiagnostic projective tests*			+							
	*Young Schema Questionnaire (YSQ-L2/SQ2)*			+							
**IQ**	*Wechsler Adult Intelligence Scale (WAIS III)*			+							
	**Terroristic assessment**										
**Radical ideas**	*Terrorist Radicalization Assessment (TRAP-18)*									+	
**Risk of violence**	*Historical Clinical Risk-20 (HCR-20)*					+					

* MINI modified version for Children and Adolescents.

## Data Availability

Upon request to the corresponding author.

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
