# Peer review of "Is There a Relationship between Psychotic Disorders and the Radicalization Process? A Systematic Review"

_medicina, 2024, doi:10.3390/medicina60060926_

Round 1

Reviewer 1 Report

Comments and Suggestions for Authors

This is a review of the manuscript titled: “Is there a relationship between psychotic disorders and the radicalization process? A systematic review”. The manuscript is well-written and provides important information about mental conditions of persons with a specific psychiatric condition. Furthermore, explore different conceptualizations about radicalization and the possibilities to participate in it.

Line 42. Please include some information about some of the theories, stages and factors to get a general idea about this complex phenomenon.

Line 53. I suggest including a flowchart that presents the evolution of radicalization and possible mechanisms to become violent as a consequence of mental disorders.

Line 154. It would be interesting for the reader if you could describe what information each of the types of studies could provide.

Line 178. Please provide general information about the causes of death of the 24% of the cases.

Line 184. The paragraph where you describe the different tools could be presented in a table and you could group the tools according to their differences.

Line 198. The radicalization process could imply different mental health conditions?

Discussion. Although you cover an important number of details, I have the impression that the relationship between mental health and radicalization is not really clear. Considering the evolution of the different conditions. Probably it would be important to mention how a person could be involved in radicalization, mentioning the process in the introduction and the effects on people with a mental health condition.

Line 250. I am not sure about the affirmation about recruiting individuals with mental disorders.

Best regards,

Author Response

R#1: This is a review of the manuscript titled: “Is there a relationship between psychotic disorders and the radicalization process? A systematic review”. The manuscript is well-written and provides important information about mental conditions of persons with a specific psychiatric condition. Furthermore, explore different conceptualizations about radicalization and the possibilities to participate in it.

A: Thank you for your positive feedback.

R#1: Line 42. Please include some information about some of the theories, stages and factors to get a general idea about this complex phenomenon.

A: Thank you for your suggestion. In the Introduction, some information about some of the theories, stages and factors to get a general idea about this complex phenomenon have been added on p.2 lines 53-68; lines 76-83).

R#1: Line 53. I suggest including a flowchart that presents the evolution of radicalization and possible mechanisms to become violent as a consequence of mental disorders.

A: Thank you for your suggestion. Based on your suggestion a flowchart figure has been added.

R#1: Line 154. It would be interesting for the reader if you could describe what information each of the types of studies could provide.

A: Thank you for your comment. As suggested, information provided from each type of study has been reported on p. 4 lines 180-186.

R#1: Line 178. Please provide general information about the causes of death of the 24% of the cases.

A: Thank you for your suggestion. Information about causes of death were reported on p. 5, lines 204-207.

R#1: Line 184. The paragraph where you describe the different tools could be presented in a table and you could group the tools according to their differences.

A: Thank you for your suggestion. A new table (named “Table 3”) has been included, summarizing the different tools identified by the literature search.

R#1: Line 198. The radicalization process could imply different mental health conditions?

A: The relationship between radicalization process and mental health conditions is complex and no direct connection has been identified by the available studies on the topic. This controversial relationship has been discussed further on p. 14, lines 305-310.

R#1: Discussion. Although you cover an important number of details, I have the impression that the relationship between mental health and radicalization is not really clear. Considering the evolution of the different conditions. Probably it would be important to mention how a person could be involved in radicalization, mentioning the process in the introduction and the effects on people with a mental health condition.

A: Thank you for this comment. We mentioned factors promoting the radicalization process in the Introduction section and further clarified in the Discussion section how individuals with mental disorders, particularly those with psychotic symptoms, are reasonably at a higher risk of being influenced by terrorist propaganda (please see on p. 2 lines 53-57, and p. 14, lines 291-301).

R#1; Line 250. I am not sure about the affirmation about recruiting individuals with mental disorders.

A: Thank you for comment, the sentence has been rephrased (p. 14 lines 281-284)

Reviewer 2 Report

Comments and Suggestions for Authors

After examining 723 published papers, the authors reviewed ten studies to try and determine whether there is a "relationship between psychotic disorders and the radicalization process." The authors cited several important limitations to their study including their reliance on observational studies and case reports which typically have a negative impact on evidence, using studies which were "limited by a high risk of bias," and studies that used extremely heterogeneous assessment tools. While research methodologies varied significantly across studies reviewed, this is to be expected in a review paper. The authors point out that others have hypothesized that mental health issues may be related to a risk of becoming radicalized. They conclude that persons with psychotic spectrum disorders "do not have a higher crime rate than the general population." That said, they cite a paper by Morris and Meloy (2020) that found "22% of a radicalized group was diagnosed with unspecified psychotic disorders." This highlights one of the problems faced by those conducting literature reviews. It may be that those most vulnerable to psychotic spectrum disorders, in particular those with Delusional disorder, may be more prone to terroristic propaganda. 

In reviewing the assessment tools that were used in papers considered for review, it is notable that only one, the MINI Neuropsychiatric Interview, provides an objective way to diagnose psychotic disorders. It is important to consider the assessment tools used when deciding whether or not to include a study in a review article. Utilizing projective tests is not generally considered the best way to diagnose psychotic disorder. Other scales used in studies that were considered included an intelligence test (WAIS III), and a suicidality scale (C-SSRS). Given that the authors did not find a relationship between psychotic disorders and the radicalization process, they did provide several useful suggestions, including using standardized assessment tools and finding better ways of limiting bias. A drawback of this study is that the final sample of studies reviewed numbered only ten (observational= 5; case reports= 4 and case series= 1). This seems like a very limited number of studies for a review paper, especially given the types of studies chosen for final review.

Comments on the Quality of English Language

The quality of English language use in this paper is generally very good. I recommend a quick read by an editor who is proficient in English.

Author Response

R#2: After examining 723 published papers, the authors reviewed ten studies to try and determine whether there is a "relationship between psychotic disorders and the radicalization process." The authors cited several important limitations to their study including their reliance on observational studies and case reports which typically have a negative impact on evidence, using studies which were "limited by a high risk of bias," and studies that used extremely heterogeneous assessment tools. While research methodologies varied significantly across studies reviewed, this is to be expected in a review paper. The authors point out that others have hypothesized that mental health issues may be related to a risk of becoming radicalized. They conclude that persons with psychotic spectrum disorders "do not have a higher crime rate than the general population." That said, they cite a paper by Morris and Meloy (2020) that found "22% of a radicalized group was diagnosed with unspecified psychotic disorders." This highlights one of the problems faced by those conducting literature reviews. It may be that those most vulnerable to psychotic spectrum disorders, in particular those with Delusional disorder, may be more prone to terroristic propaganda.

A: Thank you for your positive comments. As suggested, some tentative models explaining the higher risk of radicalization in people with psychosis have been reported on p. 14 lines 291-301.

R#2: In reviewing the assessment tools that were used in papers considered for review, it is notable that only one, the MINI Neuropsychiatric Interview, provides an objective way to diagnose psychotic disorders. It is important to consider the assessment tools used when deciding whether or not to include a study in a review article. Utilizing projective tests is not generally considered the best way to diagnose psychotic disorder. Other scales used in studies that were considered included an intelligence test (WAIS III), and a suicidality scale (C-SSRS). Given that the authors did not find a relationship between psychotic disorders and the radicalization process, they did provide several useful suggestions, including using standardized assessment tools and finding better ways of limiting bias. A drawback of this study is that the final sample of studies reviewed numbered only ten (observational= 5; case reports= 4 and case series= 1). This seems like a very limited number of studies for a review paper, especially given the types of studies chosen for final review.

A: Thank you for your suggestion. Among study’ limitations, this aspect has been acknowledged, please see on p. 15 lines 337-350.

Round 2

Reviewer 1 Report

Comments and Suggestions for Authors

Dear authors, 

Thank you for the carefully revised version of the manuscript. 

Best regards,